# Update on the Swedish Newborn Screening for Congenital Adrenal Hyperplasia Due to 21-Hydroxylase Deficiency

**DOI:** 10.3390/ijns6030071

**Published:** 2020-08-28

**Authors:** Rolf H. Zetterström, Leif Karlsson, Henrik Falhammar, Svetlana Lajic, Anna Nordenström

**Affiliations:** 1Centre for Inherited Metabolic Diseases, Karolinska University Hospital, SE-171 76 Stockholm, Sweden; rolf.zetterstrom@sll.se (R.H.Z.); leif.karlsson@sll.se (L.K.); 2Department of Molecular Medicine and Surgery, Karolinska Institutet, SE-171 76 Stockholm, Sweden; henrik.falhammar@ki.se; 3Department of Women’s and Children’s Health, Karolinska Institutet, SE-171 76 Stockholm, Sweden; svetlana.lajic@ki.se; 4Department of Endocrinology, Metabolism and Diabetes, Karolinska University Hospital, SE-171 77 Stockholm, Sweden; 5Pediatric Endocrinology Unit, Astrid Lindgren´s Children’s Hospital, Karolinska University Hospital, SE-171 76 Stockholm, Sweden

**Keywords:** neonatal screening, congenital adrenal hyperplasia, CAH, 21-hydroxylase deficiency, *CYP21A2*, dried blood spots, DBS, positive predictive value, PPV

## Abstract

Congenital adrenal hyperplasia (CAH) was the fourth disorder added to the national Swedish neonatal screening program in 1986, and approximately 115,000 newborns are screened annually. Dried blood spot (DBS) screening with measurement of 17-hydroxyprogesterone (17OHP) is also offered to older children moving to Sweden from countries lacking a national DBS screening program. Here, we report an update on the CAH screening from January 2011 until December 2019. *Results*: During the study period, 1,030,409 newborns and 34,713 older children were screened. In total, 87 newborns were verified to have CAH, which gives an overall positive predictive value (PPV) of 11% and 21% for term infants. Including the five missed CAH cases identified during this period, this gives an incidence of 1:11,200 of CAH in Sweden. Among the older children, 12 of 14 recalled cases were found to be true positive for CAH. All patients were genotyped as part of the clinical follow-up and 70% of the newborns had salt wasting (SW) CAH and 92% had classic CAH (i.e., SW and simple virilizing (SV) CAH). In the group of 12 older children, none had SW CAH and two had SV CAH. *Conclusion*: The incidence of classic CAH is relatively high in Sweden. Early genetic confirmation with *CYP21A2* genotyping has been a valuable complement to the analysis of 17OHP to predict disease severity, make treatment decisions and for the follow-up and evaluation of the screening program.

## 1. Introduction

Congenital adrenal hyperplasia (CAH) is in more than 95% of cases due to 21-hydroxylase deficiency (21OHD) caused by mutations in the *CYP21A2* gene [1,2]. It results in varying degrees of cortisol and aldosterone deficiency and at the same time an overproduction of androgens. If untreated, severe forms lead to lethal adrenal crisis in the neonatal period. Before neonatal screening, there was often a preponderance of females among patients with CAH since girls were more often identified at birth due to the prenatal virilization of external genitalia [3,4].

Measurements of 17-hydroxyprogesterone (17OHP), the metabolite before the enzymatic block, have been used as a marker for the disease [5]. Neonatal screening using filter paper cards was first accomplished in the 1970s [6]. However, neonatal screening for CAH has a relatively high false positive rate, which may have delayed its general implementation [7]. Gestational age and/or birth weight related cut-off levels have been used to try to solve this problem [8,9,10,11]. Second tier strategies using tandem mass spectrometry measuring additional steroid metabolites and using ratios of metabolites are also used and have been shown to be useful in reducing the number of false positive cases [7,12,13]. During the past decade, neonatal screening has been implemented in an increasing number of countries over the world [1].

The molecular genetics for CAH is well described [14,15]. There is a limited number of mutations, including deletion of the entire *CYP21A2* gene, that make up more than 90% of the alleles described in patients worldwide. The genotype and phenotype correlation is good, making it possible to predict the clinical severity of the disease.

Neonatal screening for CAH started in Sweden in 1986 [16]. Since then, more than 3.5 million babies have been screened. There is one national screening laboratory in Sweden managing more than 100,000 samples per year. Newborn screening is not mandatory in Sweden, but more than 99.5% of all newborns are screened. The results of the screening program between 1986 and 2011 have been reported previously [17]. Here, we extend the reporting period to also include newborns screened until 2019 and we also include the results from screening older children moving to Sweden who had not been screened for CAH in the neonatal period.

## 2. Materials and Methods

The dried blood spot (DBS) samples were collected as soon as possible after 48 h after birth (48–72 h) on Perkin Elmer 226 Ahlstrom paper (Perkin Elmer, Waltham, MA, USA). The families were given written and oral information at the time of sampling and an opt out procedure was employed. DBS screening was also offered to older children, below the age of 9 years, that moved to Sweden from countries lacking a national newborn screening program for CAH.

The 17OHP was measured using GSP^®^ instruments (Perkin Elmer, Turku, Finland). Between 2011 and 2014, AutoDELFIA^®^ was used (Perkin Elmer, Turku, Finland). Gestational age (GA) related cut-off levels were used and the cut-off level for term infants born after GA 37 weeks was 60 nmol/L in plasma and assuming a hematocrit of 50% in the blood samples. For infants born at GA 35–36, the cut-off level was 100 nmol/L, and for preterm babies born before GA 35 weeks, the cut-off level was 350 nmol/L. Children below the age of 9 years moving to Sweden were also recommended DBS screening, according to the instructions by the Swedish Board of Health and Welfare, if not screened as newborns. The cut-off level for these children was set to 50 nmol/L.

All cases with a 17OHP above the cut-off level were considered positive at the screening test and recalled for a clinical review. A case was considered true positive when diagnostic tests analyzed locally and a second DBS sample were evaluated together with the clinical assessment of the child and the diagnosis was clear. All diagnosed patients and cases with a suspected or uncertain diagnosis were genotyped for confirmation of diagnosis. Samples collected prior to 48 h of age were also analyzed when the diagnosis was suspected clinically due to virilization of the external genitalia or family history of CAH.

Genotyping was performed in one laboratory in Sweden, at the Department of Clinical Genetics, the Karolinska University Hospital in Stockholm. The whole *CYP21A2* gene was sequenced using Sanger sequencing.

The patients were clustered into genotype groups based on the severity of the milder allele. Generally, null and I2 splice, genotype groups A and B, respectively, are associated with the salt wasting (SW) phenotype. I172N is associated with SV CAH (group C), and V281L with non-classic (NC) CAH (group E). P30L results in a phenotype between SV and NC and was here defined as group D.

The study was approved by the Swedish Ethical Review Authority (approval 18 December 2019, number 2019-05816).

## 3. Results

The results from the first of January 2011 until the 31 December 2019 are reported. A total of 1,030,409 million newborn babies and 34,713 children below the age of 9 years and who had moved to Sweden were screened during this period. A summary of recalls, true positive and false negative cases are shown in Table 1. Among 791 recalled newborn babies, 87 were identified to have CAH. In Figure 1A, the percentage of recalls in the different age groups is shown.

The incidence of 21OHD in newborn babies was 1:11,200 and 92% of the detected cases had the classic form of CAH (SW and SV forms); 70% of the newborns were found to have SW CAH. The incidence of classic CAH was 1:12,300 and the NC form 1:79,300. Including the P30L in the NC genotype group gave an incidence of 1:54,200. In the group of older children, none had SW CAH, as could be expected (see below). Almost half, 48.9%, of the recalled children were born at term ≥37 weeks and 51.1% were preterm, born in or before GW 36. Older children screened when moving to Sweden represented 1.8% of recalls.

The positive predictive value (PPV) for newborns was overall 11%. For term babies, the PPV was 21%, and for preterm infants, the PPV was 1%. The false positive rate overall was 0.068%, and the false negative rate was 5.4%. The overall sensitivity was 94.6% and the specificity was 99.9%.

During the study period, more males than females were identified: 49 males and 38 females (see Figure 1B). Sixty-five percent of recalls and 56% of true positives were males. For all patients, *CYP21A2* genotyping was performed. Mutations were identified in all but one of the patients. He had markedly elevated 17OHP, >670 nmol/L in the screening, and extensive steroid hormone investigations indicated 21OHD but genetic investigations failed to identify a genetic diagnosis of adrenal enzyme deficiency thus far. The number of patients in each genotype group is shown in Figure 2A. The level of 17OHP in the screening sample versus the genotype group is shown in Figure 2B.

During the 9-year study period, five CAH patients were identified to have been missed by the screening program; see Table 1. One female (GW 34 with the I2 splice genotype (group B)) had a 17OHP value of 200 nmol/L, one female with SV CAH (GW 39) had 17OHP 39 nmol/L, one female with the P30L genotype (group D) (GW 36) had 17OHP 61 nmol/L, and two individuals with NC CAH (male GW 33 and female GW 35) had 17OHP values of 44 and 51 nmol/L, respectively.

In total, 14 children with an age between 13 months and 8 years were recalled. Twelve of these children were found to have CAH; see Table 1. Two had SV CAH, two had the P30L genotype (group D), and eight had NC CAH. In two cases, the 17OHP value normalized in the follow-up; in one child, a specific diagnosis was not identified, and in the other case, the elevated 17OHP was thought to be due to stress and malnutrition when the first sample was taken.

## 4. Discussion

This is an update on the neonatal screening for CAH in Sweden, from January 2011 until December 2019. During this period, a little more than one million newborn babies and almost 35,000 older children that moved into Sweden were screened. The majority (92%) of the patients identified in the screening had the classic form of the disease. The overall incidence of CAH in newborns in Sweden, 1:11,200, was somewhat higher than in many other countries, despite a lower incidence of NC CAH [3]. The incidence of classic CAH was 1:12,300 and of NC CAH 1:54,300.

Children moving to Sweden from countries without neonatal screening were offered screening up to the age of 8 years. Among these 35,000 children, the incidence was considerably higher, 1:2900. Two patients had the classic SV form of CAH and would definitely have benefited from earlier diagnosis. The others belonged to the genotype groups P30L and NC CAH, with no risk of salt-losing crisis [18]. The overall aim of the screening is to prevent neonatal salt-losing crises and death. However, our impression is that some of the children with NC CAH benefit from being detected early. Treatment with hydrocortisone has only been initiated if they had clinical symptoms, markedly accelerated bone age or a stimulated cortisol level that was considered insufficient for age. In some cases, the families were advised to give stress doses of hydrocortisone in case of fever without starting regular treatment. It is known that some individuals with the NC genotype develop more symptoms and have an insufficient maximum capacity to produce cortisol compared to others, despite the same *CYP21A2* genotype (18). In our experience, the children with NC CAH identified in the neonatal screening did not seldom require treatment, suggesting that being identified in the neonatal screening is an indication that they have a genetic background that predisposes them to produce more adrenal androgens and hence develop symptoms.

The efficiency of the screening program depends on the samples being collected early enough to avoid salt-losing crisis [17]. The earlier time of sampling, from 72 to 120 h to as soon as possible after 48 h, implemented prior to the extension of the Swedish national screening program in 2010, was therefore positive also for the screening for CAH. The turnaround time and the feedback to the health professionals regarding an elevated sample are all important steps for an efficient screening program (see review article in this issue).

The earlier time of sampling may have affected the distribution of males and females, with 65% of recalls and 56% of true positives being males. In our previous study, the males constituted 61.5% and females 38.5% of recalls but the number of true positive cases was similar between males and females [17]. The PPV was somewhat lower in boys than in girls. Interestingly, four of the missed cases were females, indicating a somewhat lower overall sensitivity for females or, more likely, the girls were more often identified clinically due to signs of increased androgen production when they sought medical attention. However, there was no significant sex difference in the sensitivity, with 87.2% for boys and 81.6% for the girls, despite a PPV for boys of 11% and for girls of 17%, as we previously reported [17].

The aim of the newborn screening for CAH is to avoid salt-losing crisis and neonatal death. The false negative cases among infants born full-term have most often had less severe forms of CAH, the I172N and the V281L genotype groups, with low or no risk of developing neonatal salt-losing crisis. Over the years, four patients in the I2 splice genotype group have been missed by the screening, three reported previously [17]. They were diagnosed later during childhood but none of them had developed any adrenal crisis. The one reported here, however, had an uncertain gestational age and other medical issues requiring intervention. It is known that the genotype and phenotype correlation is especially difficult for individuals in the I2 splice genotype group. A few patients with this genotype have been reported not to develop salt crises and there are individuals who were identified when investigated as presumed carriers but never developed any symptoms of cortisol deficiency or increased adrenal androgen production [3,19].

Further improvement in screening efficiency and outcome will require implementation of a second tier. Between September 2011 and May 2013, we performed second tier tandem mass spectrometry analysis on term children with 17OHP values between 60 and 100 nmol/L, measuring 17OHP, androstenedione, cortisol, 21-deoxycortisol and 11-deoxycortisol [20]. Work is now in progress to implement MS/MS second tier testing as a routine procedure and to start using the Collaborative Laboratory Integrated Reports (see www.clir.mayo.edu and Special Issue in this journal).

## 5. Conclusions

The newborn screening for CAH in Sweden had an incidence of 1:11,200 and the present overall PPV was 11% and for full term infants was 21%. Screening of older children moving to Sweden from countries lacking DBS screening for CAH has also proven to be valuable. All infants identified were genotyped, which enabled classification of the disease severity. This lends valuable support in clinical decision-making concerning treatment and facilitates assessment of the efficacy of the screening program.

## Figures and Tables

**Figure 1 IJNS-06-00071-f001:**
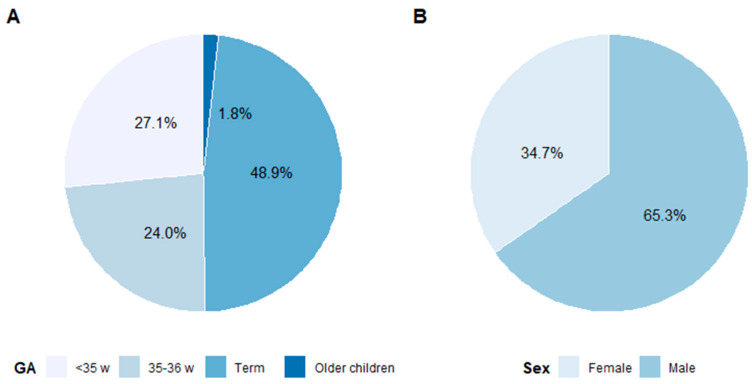
Gestational age (**A**) and sex (**B**) distribution of all recalls in the neonatal screening program for CAH in Sweden during the study period.

**Figure 2 IJNS-06-00071-f002:**
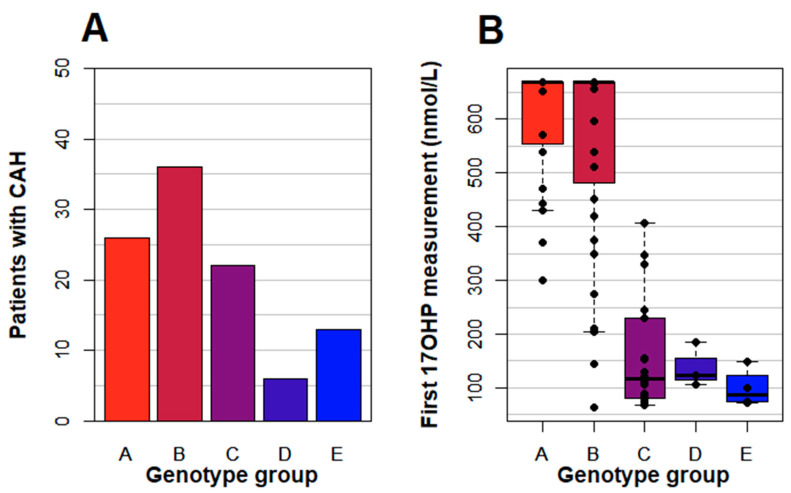
Distribution of all CAH patients’ *CYP21A2* genotype groups (*n* = 103) (**A**) and correlations of the 17OHP values in the neonatal screening samples to the *CYP21A2* genotype groups (**B**). The 17OHP level is given up to 670 nmol/L by the laboratory, which means that higher levels are reported as >670 nmol/L. Five patients are not included in graph B due to a missing genotype in one case, missing 17OHP in one case and sampling prior to 48 h in three cases.

**Table 1 IJNS-06-00071-t001:** All recalls, true positive cases, false negative cases and PPV values for the CAH screening in Sweden of more than one million children between 2011 and 2019.

Age Group	Total Recalls	True Positive Cases	False Negative Cases	PPV
**GA < 35 w**	214	0	2	-
**GA 35–36 w**	189	4	2	2%
**Term**	388	83	1	22%
**Total newborns**	791	87	5	11%
**Older children**	14	12	-	86%

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
