# Peer review of "Update on the Swedish Newborn Screening for Congenital Adrenal Hyperplasia Due to 21-Hydroxylase Deficiency"

_2409-515X, 2020, doi:10.3390/ijns6030071_

Round 1

Reviewer 1 Report

  1. Materials and methods, lines 79-83: Please explain the exact method used for CYP21A2 genotyping. Was it by an allele-specific panel, or was sequencing carried out?
  2. Results, lines 112-113: How extensive was the search for mutation in the one patient for whom no mutations could be identified?
  3. Results and Discussion: It would be helpful to separate the incidence of classic CAH from nonclassic CAH, since most newborn screening programs have their primary target set for classic CAH.
  4. Discussion, line 137: The authors state that there is a lower incidence of nonclassic CAH in Sweden compared with other countries. Please state the incidence.
  5. Discussion: This reviewer is interested in knowing the criteria the authors would put forth for benefiting children with Val-281-Leu mutations by detection in infancy, as many of these children apparently need not seek medical treatment for their mild enzyme deficiency.

Author Response

Thank you for the valuable suggestions: 

1. Materials and methods, lines 79-83: Please explain the exact method used for CYP21A2genotyping. Was it by an allele-specific panel, or was sequencing carried out?

Response: This information has now been added.

2. Results, lines 112-113: How extensive was the search for mutation in the one patient for whom no mutations could be identified?

Response: A short information has now been added.

3. Results and Discussion: It would be helpful to separate the incidence of classic CAH from nonclassic CAH, since most newborn screening programs have their primary target set for classic CAH.

 Response: Incidence figures are now given for NC and Classic CAH separate and combined

4. Discussion, line 137: The authors state that there is a lower incidence of nonclassic CAH in Sweden compared with other countries. Please state the incidence.

Response: This information is given in our previous publication, Gidlöf et al 2013. This is now referenced in the text.

5. Discussion: This reviewer is interested in knowing the criteria the authors would put forth for benefiting children with Val-281-Leu mutations by detection in infancy, as many of these children apparently need not seek medical treatment for their mild enzyme deficiency.

Response: A paragraph discussing this has now been added to the discussion.

Reviewer 2 Report

Thank you for giving me the opportunity to review this paper. The paper continues the significant contribution to the screening literature on CAH by the Swedish newborn screening programme.  I have the following comments on the manuscript:

I would suggest that the screening target be defined and then reflected in the results and discussion. It appears that non-classical CAH is a target for screening as 2 cases detected after the newborn period were stated as been missed by screening (line 125). In a previous report, GidlÓ§f et al (Lancet Diabetes Endocrinol 2013;1:35-42) states that the “programme does not try to detect milder forms of the disorder [CAH]”. It would be useful for the literature if this could be clarified in the introduction and the results (false negatives = 3, sensitivity = 96.7%)

It is not clear from the introduction what is meant by a positive screening test. On line 78 it is suggested that an elevated 17OHP is considered a recall and a test is considered positive when diagnostic tests and CYP21A2 genotyping and second sample are positive. However, in the results section, PPV values suggest that recalls are all considered positive. I suggest elevated 17OHP test results are considered positive and that 704 screening results are considered false positives. The definition of line 78-79 would then read that a “test is considered true positive when diagnostic test analysed locally…..”

It would be useful to clarify in the methods section if all neonates with an elevated initial 17OHP were genotyped and clinically reviewed. The method section (lines 78-79) suggested they were. However, on line 112 it is stated that “For all patients CYP21A2 genotyping was performed. Mutations were identified in all but one patient, despite a markedly elevated 17OHP” This is inconsistent with the method. Were all cases clinically diagnosed and then genotyped? Also was the baby with a marked elevated 17OHP diagnosed clinically but considered a false positive because of genotyping. The screening and clinical out come (?another form of CAH) would be useful information to include.

Line 137/138. It is stated the “over all incidence of CAH in Sweden is 1:11200…….despite a lower incidence of NC-CAH”. The results suggest that NC-CAH incidence is much higher based on the screening results on older children (8 out of 34713, 1:4339).

The paragraph with lines 145-150 should be re-written. This paragraph seems to be about the importance of early detection and clinical referral and treatment.

It would be useful to compare the positive/true positive screening results between male and females using a statistical tool (e.g chi-squared test) and compare with other reports. E.g.  difference between male and female 17OHP (Pearce et al. Molecular genetics and Metabolism Reports Reports 2016; 7:1-7).

Lines 166-169. It is stated that the genotype to phenotype correlation is difficult to predict with the I2 splice genotype. However, the most common genotype was I2 splice genotype (figure 2). In line 55 it is stated that the clinical severity of genotype can be predicted. These are conflicting statements

I have also noted the following typographical errors

line 23 - remove the word "results" unless it is a heading in the abstract

Figure 1 line 107. Missing legend for 27.1% (assuming term neonates). Age should read Gestational Age

Author Response

We thank the reviewer for valuable suggestions:

Thank you for giving me the opportunity to review this paper. The paper continues the significant contribution to the screening literature on CAH by the Swedish newborn screening programme.  I have the following comments on the manuscript:

I would suggest that the screening target be defined and then reflected in the results and discussion. It appears that non-classical CAH is a target for screening as 2 cases detected after the newborn period were stated as been missed by screening (line 125). In a previous report, GidlÓ§f et al (Lancet Diabetes Endocrinol 2013;1:35-42) states that the “programme does not try to detect milder forms of the disorder [CAH]”. It would be useful for the literature if this could be clarified in the introduction and the results (false negatives = 3, sensitivity = 96.7%)

Response: This information is now more explicitly discussed in the discussion.

It is not clear from the introduction what is meant by a positive screening test. On line 78 it is suggested that an elevated 17OHP is considered a recall and a test is considered positive when diagnostic tests and CYP21A2 genotyping and second sample are positive. However, in the results section, PPV values suggest that recalls are all considered positive. I suggest elevated 17OHP test results are considered positive and that 704 screening results are considered false positives. The definition of line 78-79 would then read that a “test is considered true positive when diagnostic test analysed locally…..”

Response: The sentence describing this has now been rewritten to make this more clear

It would be useful to clarify in the methods section if all neonates with an elevated initial 17OHP were genotyped and clinically reviewed. The method section (lines 78-79) suggested they were. However, on line 112 it is stated that “For all patients CYP21A2 genotyping was performed. Mutations were identified in all but one patient, despite a markedly elevated 17OHP” This is inconsistent with the method. Were all cases clinically diagnosed and then genotyped? Also was the baby with a marked elevated 17OHP diagnosed clinically but considered a false positive because of genotyping. The screening and clinical out come (?another form of CAH) would be useful information to include.

Response: All cases suspected from hormonal investigations and with suspected or unclear diagnosis are genotyped. Patients with other forms of CAH were not included in this report. We have now tried to make this more clear.

Line 137/138. It is stated the “over all incidence of CAH in Sweden is 1:11200…….despite a lower incidence of NC-CAH”. The results suggest that NC-CAH incidence is much higher based on the screening results on older children (8 out of 34713, 1:4339).

Response: The information on incidence is given in our previous publication, Gidlöf et al 2013. This is now referenced in the text.

 The paragraph with lines 145-150 should be re-written. This paragraph seems to be about the importance of early detection and clinical referral and treatment.

Response: The sentence has now been re-written as suggested.

 It would be useful to compare the positive/true positive screening results between male and females using a statistical tool (e.g chi-squared test) and compare with other reports. E.g.  difference between male and female 17OHP (Pearce et al. Molecular genetics and Metabolism Reports Reports 2016; 7:1-7).

Response: This was calculated in our previous publication Gidlöf 2013, but is not easily done here since the exact number of boys and girls born in the population at this time was not available. A reference to the previous information is made in the text.

Lines 166-169. It is stated that the genotype to phenotype correlation is difficult to predict with the I2 splice genotype. However, the most common genotype was I2 splice genotype (figure 2). In line 55 it is stated that the clinical severity of genotype can be predicted. These are conflicting statements

Response: A more extensive explanation for this statement is now given in the text in the discussion.

I have also noted the following typographical errors

line 23 - remove the word "results" unless it is a heading in the abstract

Response: This is a heading – now in italics to make this clear

Figure 1 line 107. Missing legend for 27.1% (assuming term neonates). Age should read Gestational Age

Response: Thank you for noting this. The Figure has now been corrected.